# Tolerization with a Novel Dual-Acting Liposomal Tim Agonist Prepares the Immune System for the Success of Gene Therapy

**DOI:** 10.3390/ijms26083830

**Published:** 2025-04-18

**Authors:** Abdulraouf Ramadan, Pushpa Rao, Saleh Allababidi, Raed Khashan, Anas M. Fathallah

**Affiliations:** 1LAPIX Therapeutics Inc., Cambridge, MA 02141, USA; 2Artelligence Therapeutics LLC, Philadelphia, PA 19114, USA; raed.khashan@gmail.com

**Keywords:** gene therapy, immunogenicity, tolerance induction

## Abstract

Gene therapy holds great promise for treating various congenital rare diseases. However, immunogenicity against viral vectors used in gene therapy remains a challenge, impacting both the safety and efficacy of gene therapy products. Neutralizing antibodies against the vector capsid proteins impact the ability to re-dose patients, which a growing body of evidence suggests might be required for some indications and certain younger patient populations. In this communication, we report a novel dual-acting liposomal formulation that induces immune tolerance toward adeno-associated virus 9^null^ (AAV9^null^) capsid proteins. We present in silico data on our first- and second-generation Tim agonist molecules as well as in vitro and in vivo data supporting the generation of antigen-specific regulatory T cells (Tregs) as well as abrogation of antibody response to AAV9^null^ capsid in our animal models. These early data are encouraging and may offer a new solution to mitigate the immunogenicity induced by gene therapy products.

## 1. Introduction

Since its clinical trial debut thirty years ago, more than 1700 gene therapy clinical trials have been conducted worldwide [1], and 9 US-FDA approvals [2] have been granted. The great promise of gene therapy has been hindered by safety, tolerability, and durability issues [3]. Viral vectors comprising a protein or lipid-protein capsid and genetic payload are the preferred platform for gene therapy. Lentiviral vectors (LV), Adenoviral vectors (AV), and Adeno-associated viral vectors (AAV) are the most used gene transfer vectors in clinical trials. While still in use, Lentiviral fell out of favor after a meta-analysis revealed the association between LVs and genotoxicity [4]. AVs have high transduction efficiency and persistence in the host cells but are associated with high pre-existing immunity in humans, leading to life-threatening strong innate immune responses to its capsid proteins [5]. AVs, however, found utility in the vaccine space. AAV vectors have been universally recognized as versatile vectors for gene therapy. While AAVs are considered the least immunogenic compared to other vectors, AAVs still induce a humoral (i.e., anti-AAV antibodies) and cell-mediated (i.e., effector cells) immunogenicity-related concerns against immunogenic components of the AAV-based gene therapy products.

Neutralizing antibody responses against the AAV capsid protein had limited the ability to re-dose patients when needed. Such needs for re-dosing exist due to the poor predictability of preclinical efficacious dose extrapolations, loss of therapeutic durability due to potential cytotoxic T cell responses against transduced cells (particularly in the liver) [6,7,8], and the generally non-integrating episomal nature of AAV genomes that results in progressive loss of transduction in proliferating tissues such as livers in pediatric populations [9]. The relative safety profile of AAV, compared to other vectors, is, unfortunately, countered by the low, stable transduction efficacy, which necessitates re-dosing. Indeed, clinical experience with the development of gene therapies for spinal muscular atrophy (SMA) and hemophilia suggests that transduction is not permanent, and repeat dosing is required [10]. However, a second dose is not possible since the immunogenicity developed against the capsid will generate neutralizing antibodies and increase the risk of uncontrolled and potentially lethal immune response [10,11]. Immunogenicity to viral vectors is a major reason why pre-existing titers to the viral vector are contraindicated in gene therapy clinical trials [10,12].

For example, 100% of patients treated with the approved gene therapy for SMA, Zolgensma, developed an immune response, essentially eliminating the possibility of repeated dosing [12]. Due to the inability to re-dose patients with Zolgensma, a recent clinical trial was initiated to assess the benefit of SPINRAZA (an antisense oligonucleotides therapy) in SMA patients who received Zolgensma but could benefit from the additional somatic motor neuron protein provided by SPINRAZA [13]. Hence, a critical unmet need exists for an approach to mitigate the immunogenicity of gene therapy.

In this report, we present data on a novel dual-acting liposomal formulation targeting the Tim family of receptors to induce CD4+/Foxp3+/CTLA4+/ICOS^lo^ tolerogenic regulatory T cells (Tregs). First, we present data on LPX1, a proposed Tim-family agonist that was selected from several early candidates based on the rigid in silico binding model to Tim bound to its natural ligand. Then, we present data on the second generation of Tim-family agonists, LPX2 and LPX3, with higher in silico binding affinity to Tim. LPX2 works independently from any carrier, and LPX3 is designed to be part of a lipid membrane of the liposome that will incorporate the AAV9^null^ (Figure 1); we formulated both in a dual-acting LPX-TI that can generate highly functional antigen-specific Treg.

## 2. Results

### 2.1. LPX1 Induces Tolerance Without Altering Innate Immunity

#### 2.1.1. Immunogenicity of AAV9^null^ Administered with LPX1

A total of 8/8 animals in the AAV9^null^ SC control group developed anti-AAV9 antibodies vs. 0/8 in the AAV9^null^+LPX1 SC group (Figure 2A). FoxP3+/CD4+ Tregs increased by 122% in AAV9^null^+LPX1- vs. AAV9^null^-treated mice (Figure 2B).

#### 2.1.2. The Effects of LPX1 on Innate Immunity

Both dexamethasone and rapamycin abrogated the splenocyte’s TNF-α response to the TLR2-ligand PGN, TLR4-ligand LPS, and TLR9-ligand ODN-CPG regardless of the dose of ligand (Figure 2C). Treatment with LPX1 resulted in limited inhibition of TNF-α response compared with dexamethasone and rapamycin. There was a statistically significant reduction in TNF-α in the presence of low-dose PGN (1 ng/mL) only, but not at higher PGN doses (Figure 2C). Similarly, LPX1 resulted in a relatively limited inhibition of splenocyte’s response to LPS; however, a statistically significant reduction in TNF-α in the mid and high LPS dose group was observed (Figure 2C).

Treatment with dexamethasone, rapamycin, or LPX1 abrogated splenocyte’s response to the TLR3-ligand PolyI:C at all doses vs. control. Treatment with dexamethasone, rapamycin, or LPX1 reduced the splenocyte’s response to a high dose of the TLR7-ligand Poly-U vs. control (statistically significant) (Figure 2C). At the medium and low doses of Poly-U, dexamethasone, rapamycin, and LPX1 resulted in an equivalent response to control (Figure 2C). Finally, Treatment with dexamethasone, rapamycin, or LPX1 resulted in a statistically significant reduction in the splenocyte’s response to a medium dose of the TLR9-ligand ODN CpG vs. control (Figure 2C). At high ODN-CpG doses, dexamethasone and rapamycin suppressed splenocytes’ response but not LPX1’s (Figure 2C).

### 2.2. Improved Tim Binding of Second-Generation LPX Compounds

#### 2.2.1. Validation of Docking Model

Redocking the natural ligand resulted in poses that matched the native binding configuration with a root mean square deviation (RMSD) of 2.213 Å and ranked among the top 11 poses, demonstrating the reliability of the docking procedure. The computationally calculated binding affinity of PS to its protein was found to be −6.0 kcal/mol.

#### 2.2.2. Docking New Drug Candidates

In silico binding affinity values of rigid docking for the top 10 binding poses showed that LPX1 binding was ~1 kcal/mol lower than Tim’s natural ligands headgroup OPLS. LPX2 binding was ~2.2 kcal/mol lower than OPLS and 1.8 kcal/mol lower than the first-generation LPX1. LPX3, the lipid intercalating compound, has a binding that is comparable to the natural ligand OPLS (Table 1).

### 2.3. LPX-TI Induces AAV9^null^-Dose-Dependent Foxp3+ Upregulation In Vivo

Mice treated with 10^7^ particles/mL AAV9^null^-LPX-TI SC for 5 days showed a statistically significant increase in Foxp3+/CD4+ T-cell (mean = 8.5, SD = 1.4) vs. mice dosed with AAV9^null^ (mean = 3.4, SD = 1.1) or with higher doses (10^8^ or 10^9^ particles/mL) of AAV9^null^-LPX-TI (mean = 6.0, SD = 0.8, mean = 4.8. SD = 0.4, respectively) (Figure 3A).

### 2.4. IM Administration of AAV9^null^-LPX-TI Induces Highly Tolerogenic Tregs

There were no statistical differences in Foxp3+/CD4+ Tregs between mice administered 10^7^ AAV9^null^ IM with or without LPX-TI IM or SC (Figure 3B). Mice treated with 10^7^ particles/mL AAV9^null^-LPX-TI IM, however, had a statistically significant increase in CD4+/Foxp3+/CTLA4+ Tregs (mean = 4.8, SD = 1.4) vs. mice administered 10^7^/mL AAV9^null^ or 10^7^ particles/mL AAV9^null^-LPX-TI SC (mean = 2.4, SD = 0.1, mean = 2.5, SD = 0.2 respectively) (Figure 3C).

Mice treated with 10^7^ particles/mL AAV9^null^-LPX-TI IM or SC had statistically lower levels of CD4+/Foxp3+/ICOS+ (non-tolerogenic Tregs) (mean = 2.7, SD = 0.7 and mean = 2.5, SD = 0.3, respectively) vs. mice treated with 10^7^/mL AAV9^null^ alone IM (Figure 3D).

### 2.5. AAV9^null^-LPX-TI Expands Antigen-Specific Tregs in a Recall Response

A statistically significant and dose-dependent expansion of antigen-specific CD4+/Foxp3+ Tregs was observed in splenocytes of animals treated with AAV9^null^-LPX-TI IM at all doses in vitro vs. animals treated with AAV9^null^ IM (Figure 3D).

Splenocytes from animals treated with AAV9^null^-LPX-TI SC had a statistically significant expansion of antigen-specific CD4+/Foxp3+ T-cells in response to 10^6^ and 10^7^ particles/mL vs. splenocytes from AAV9^null^ treated animals (Figure 3D). When comparing AAV9^null^/LPX-TI IM vs. SC, antigen-specific CD4+/Foxp3+ T-cells expansion was statistically higher in splenocytes from IM-dosed mice vs. SC at 10^6^ and 10^7^ AAV9/mL (Figure 3D).

### 2.6. IM Administration of AAV9^null^-LPX-TI Did Not Produce Anti-AAV9 Antibody Titer

All animals in the AAV9^null^ group developed total anti-AAV9 antibody titer vs. none in the AAV9^null^-LPX-TI IM group at 1:20 and 1:40 dilution levels. The AAV9^null^-LPX-TI SC group had mixed results at both dilution levels (Figure 4).

## 3. Discussion

Antigen-specific immune tolerance holds great promise to mitigate the immunogenicity of gene therapy and allow for repeated dosing. To date, antigen-specific tolerance induction has depended on two approaches: (1) cell therapy such as TolDCs [14] or (2) nanoparticles-carried antigen/antigenic fragments along with an immunosuppressive agent (ex Rapamycin [15]) or immunomodulating lipids (ex PS) containing liposomes [16]. Tolerizing using immunosuppressive molecules leads to antigen presentation without co-stimulation, resulting in T-cell anergy and/or expansion of inducible Tregs (iTregs) [17]. iTregs are known to be unstable and can change their function depending on the environment [18]. More recently, the addition of IL-2, or IL-2 agonists, is being explored to aid in antigen-specific tolerance. While the stability of Tregs generated by this approach is still being evaluated, data suggest Tregs induced by IL-2 may not be stable [19].

The use of lipids (PS/PS-derivatives) can induce n-Tregs, which are more stable than iTregs [20]. However, PS’s low affinity to Tim3 vs. other Tims [21] poses a challenge since Tim3 is a key player in immune tolerance [22]. PS-mediated tolerance usually results in a reduction, but not elimination, of the immune response [15]. Finally, tolerating an antigenic epitope fragment is inherently problematic as it does not account for all possible linear epitopes nor structural antigenic epitopes that can be recognized by BCR. However, tolerizing with intact capsid (protein) generates a pool of polyclonal antigen-specific T and B cells covering all possible immunogenic linear and structural epitopes [23]

We have demonstrated that our first-generation LPX1 Tim-agonist induces tolerance, as evident by enhanced Treg expression and lack/reduction of anti-AAV9 titers in mice treated with LPX1 with AAV9^null^ vs. AAV9^null^ alone at the same frequency and route (Figure 1). Furthermore, LPX1, unlike dexamethasone or rapamycin, inhibited intracellular TLRs (TLR3 and 7) but not TLR2, 4, and 9, suggesting that LPX1 is not a general immunosuppressor but rather a selective immune tolerance agent.

Second-generation Tim-agonists were developed to lower the effective dose and ensure both local and systemic engagement of immune cells for antigen-specific presentation. The reduction in ΔG predicted by our in-silico model suggests at least 10 times better affinity for LPX2 vs. LPX1 and almost 100 times better affinity compared to Tim’s natural ligand’s headgroup OPLS. The enhanced binding addresses the issue of poor interaction between Tim and its natural ligand, allowing us to reduce in vivo dose from 5 mg for LPX1 to 100/150 ug for LPX2/LPX3. Liposomal formulation encapsulating AAV9^null^ was used to address the second objective. Encapsulation conceals the viral capsid from TLR2 [24] and ensures local exposure to the immune system at the injection site. The hydration buffer contained LPX2 to engage immune cells systemically. The resulting formulation is referred to as AAV9^null^-LPX-TI.

When we tested different AAV9^null^-LPX-TI doses SC in mice, we observed an inverse relationship between the AAV9^null^ dose and Foxp3+/CD4+ T-cells. We found that 10^7^ AAV9^null^ particles/mL LPX-TI resulted in a statistically significant increase in Foxp3+/CD4+ Tregs vs. higher AAV9^null^ to LPX-TI ratios or AAV9^null^ control group (Figure 3A). Since not all Tregs are created equal, we evaluated the functionality of those Tregs after SC and IM administration of AAV9^null^-LPX-TI. We observed a higher-than-expected production of Foxp3+/CD4+ Tregs in animals treated with five daily IM AAV9^null^ doses (Figure 3B); however, those Tregs were also ICOS+/CTLA4- suggesting they are suppressive, but non-tolerogenic Tregs (Figure 3C,D) [25]. IM administration of AAV9^null^-LPX-TI upregulated tolerogenic Foxp3+/CTLA4+/ICOS^lo^ Tregs (Figure 3C,D). SC administration of AAV9^null^-LPX-TI resulted in statistically lower ICOS+ Tregs compared to the AAV9^null^ alone group and statistically lower CTLA4+ Tregs compared to the IM AAV9^null^-LPX-TI group. These data suggest that the IM route could be better than SC for tolerance induction. The data from the recall response study supports this notion. Indeed, splenocytes isolated from mice administered IM or SC with AAV9^null^-LPX-TI had a statistically higher expansion of Tregs compared to mice administered AAV9^null^ alone IM (Figure 3E). Similarly, Treg expansion was statistically higher in mice that were administered IM vs. SC AAV9^null^-LPX-TI.

Finally, titer analysis supports the finding of our ex vivo T-cell phenotyping and recall responses. As seen in Figure 4, mice treated with five daily doses of AAV9^null^-LPX-TI IM did not generate anti-AAV9 antibodies vs. mice treated with five daily IM doses of AAV9^null^ alone. Mice treated with five daily AAV9^null^-LPX-TI doses SC had a mixed anti-AAV9 titer response.

Taken together, the data presented in this communication suggest that IM administration of AAV9-LPX-TI creates a tolerogenic environment and upregulates antigen-specific tolerogenic Foxp3+/CTLA4+/ICOS^lo^ Tregs that, upon re-exposure to AAV9, induced a tolerogenic response. We propose two possible scenarios for the clinical application of this approach: First, induce immune tolerance to therapeutic viral vector by pre-treating the subject with null virus matching its serotype before administering the therapeutics dose at the intended route and dose (Figure 5A). This aligns with data presented here using the AAV9^null^ virus. Second, the virus containing a genetic payload could be co-formulated with LPX-TI to induce tolerance upon administration (Figure 5B) with or without pre-tolerization. We did not evaluate this option in the current report; however, we plan to address this in future studies designed to evaluate both tolerization towards the envelope/capsid as well as the transduction efficacy of gene therapy coformulation with LPX-TI. Together, our promising data may offer a possible solution to the immunogenicity of gene therapy. The suggested approach of using LPX-TI to induce tolerance towards encapsulated virus could be extended for other viral vectors that might have better transduction efficiency and stability but have been abandoned due to their high immunogenicity risk. Further characterization and optimization of this approach is warranted.

## 4. Materials and Methods

**Docking Simulations**: AutoDock Vina (ADVina) version 1.1.2 was used to perform docking simulations in order to assess the binding affinities of several proposed molecular structures to the Tim receptor.

**Preparation for Docking**: To prepare for docking, the co-crystal structure of the Tim- protein bound to its endogenous ligand, dicaproyl-phosphatidylserine (PS), was downloaded from the Protein Data Bank (PDB ID: 3BIB). The ligand (PS) was removed from the binding site to allow docking of the proposed molecules. All water molecules were removed except for one, which was essential for forming a hydrogen bond bridge between PS and the protein. Charges for all atoms in the protein were calculated at a physiological pH of 7.4, matching the conditions used in the in vitro experiments. Similarly, charges for all docked molecules were also computed at the same physiological pH. Docking was performed with both rigid and flexible protocols, and the top 20 binding poses for each molecule were identified for further analysis.

**Validation of the Docking Method**: To validate the docking approach, the endogenous ligand (PS) was re-docked into the protein’s binding site after its removal. The root mean square deviation (RMSD) of the pose that matched the native configuration was calculated and ranked to demonstrate the reliability of the docking procedure.

**Docking of Experimentally Tested and Proposed Structures**: After validating the docking method, several compounds—OPLS, LPX1, LPX2, and LPX3—were docked into the Tim receptor’s binding site using both rigid and flexible docking methods. These results were compared with experimental data. Mice: seven-to-ten-week-old, female C57BL/6 mice were obtained from Charle River Laboratories. All mice were housed under specific pathogen-free conditions at the Tufts University/Tufts Medical Center.

**In vitro TLR stimulation**: 2 × 10^5^ splenocytes/well cultured in DMEM supplemented with 10% FBS, 2 mM l-glutamine, 1% penicillin/streptomycin, 1 mM sodium pyruvate, and 50 µM β-mercaptoethanol (Life Technologies, Carlsbad, CA, USA) as described in [26]. Splenocytes were preincubated with either 10 µg/mL of LPX-TI1, 10 ug/mL of rapamycin (Sigma-Aldrich, St. Louis, MO, USA), or 50 ng/mL dexamethasone (Sigma, Sigma-Aldrich, St. Louis, MO, USA). Then, cells were stimulated with three ascending doses of each TLR agonist at 37 °C for 48 h. The supernatant was collected for TNFα measurement by ELISA (R&D Systems, Minneapolis, MN, USA).

**AAV9^null^ was sourced** from Vigene Biosciences (Rockville, MD, USA) catalog number RS-AAV9-ET. The physical particle concentration, as determined by ELSA, was 1.79 × 10^12^ VP/mL. The purity, size, and stoichiometry of the viral protein were determined by reduced SDS-PAGE. Sourced AAV9^null^ was negative for mycoplasma and had an endotoxin level <10 EU/mL as determined by the LAL test. All tests were performed by Vigene Biosciences per their SOPs.

**Liposomal formulations**: Liposomes were prepared by the thin film method. Briefly, LPX3 was dissolved in chloroform and mixed with DMPC (10:90 mole/mole) in a round-bottom flask. The thin film was formed using a rotavap. The films were hydrated with buffer containing LPX2 and different concentrations of AAV9^null^ to give a final AAV9^null^ dose of 10^8^, 10^9^, or 10^10^ particles/mL. The resulting formulation is referred to throughout as LPX-TI.

**Immunization**: Mice (n = 8 per group) were immunized SC with 10^6^ AAV9^null^ particles alone or with LPX1 once a week for 4 weeks, and blood samples were collected for titer analysis 2 weeks after the last dose. Alternatively, mice (n = 4 per group) were immunized with AAV9^null^ alone or formulated with LPX-TI liposome IM or SC once a day and 5 days. Blood samples and spleens were collected five days after the last dose. Titer analysis was performed as per [27].

**Recall response**: Spleens from treated mice were collected, and a single-cell suspension was cultured in DMEM supplemented with 10% FBS, 2 mM l-glutamine, 1% penicillin/streptomycin, 1 mM sodium pyruvate, and 50 µM β-mercaptoethanol (Life Technologies) in the presence of ascending doses of AAV9^null^ for five days.

**Flow cytometry staining**: In vitro or isolated from the spleen ex vivo were washed and preincubated with purified anti-mouse CD16/CD32 mAb for 10–20 min at 4°C to prevent nonspecific binding of the antibodies. The cells were subsequently incubated for 30 min at 4 °C with antibodies for surface staining anti-CD3 and anti-CD4. Fixable viability dye was used to distinguish live. Transcription factor fixation and permeabilization kit were used for intracellular staining of Foxp3 and CTLA4.

**Statistics**: For titer analysis in Figure 2A, an unpaired, non-parametric *t*-test (Mann–Whitney test) was used. A 2-way ANOVA with multiple comparisons to the control was used for Figure 2C and Figure 3E. Everywhere else, one-way ANOVA was used. All statistical analyses were performed using GraphPad Prism 10.

## 5. Conclusions

Together, our promising data may offer a solution to the immunogenicity of gene therapy. The suggested approach of using LPX-TI to induce tolerance towards encapsulated cargo could be extended for other viral vectors that might have better transduction efficiency and stability but have been abandoned due to their high immunogenicity risk. Further characterization and optimization of this approach is warranted.

## Figures and Tables

**Figure 1 ijms-26-03830-f001:**
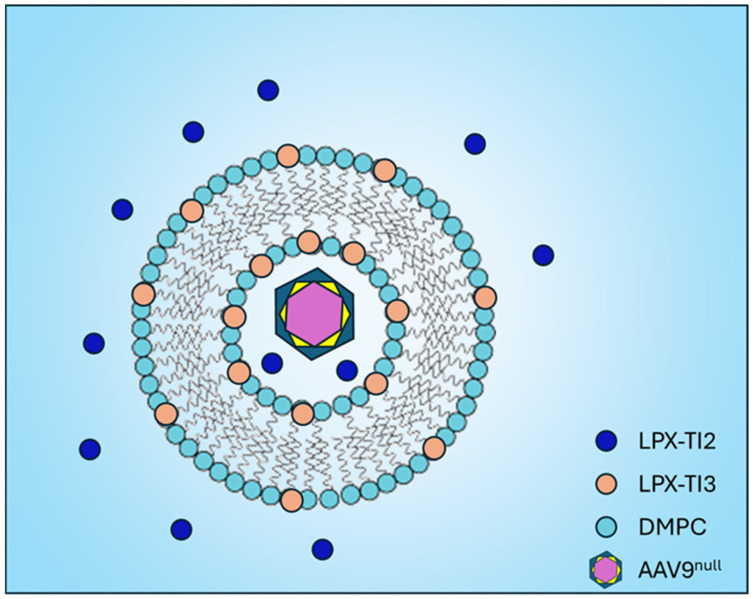
A schema demonstrating AAV9^null^ encapsulated in LPX-TI liposomes, with LPX3 in the lipid bilayer and LPX2 in the hydration buffer.

**Figure 2 ijms-26-03830-f002:**
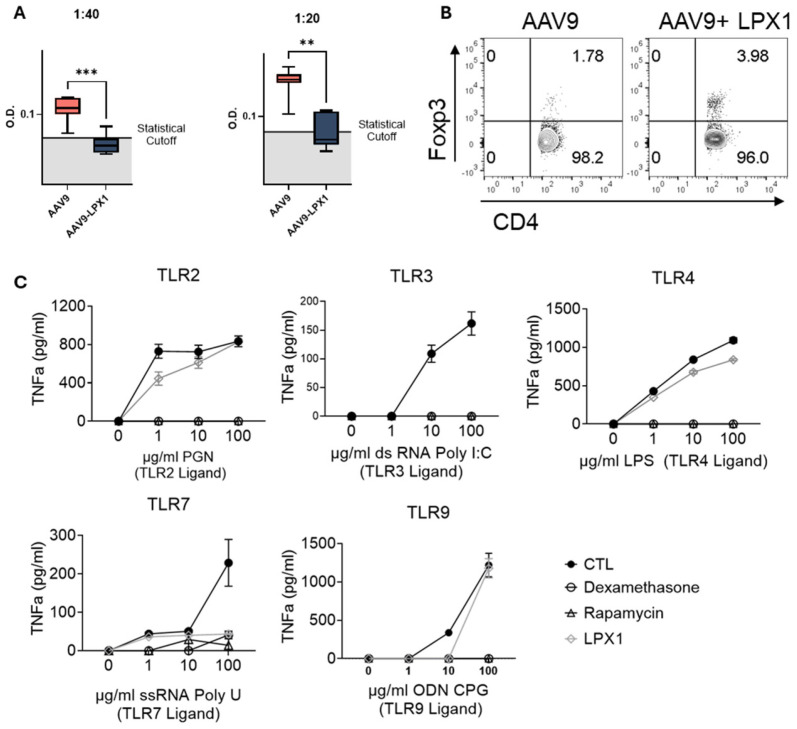
(**A**) Anti-AAV9 antibody levels measured at the end of the treatment period. (**B**) Flow cytometry analysis of Tregs from splenocytes of animals treated with LPX-TI1 or control, assessed based on the route of administration using ELISA. (**C**) TNF-α secretion from splenocytes pretreated with dexamethasone, rapamycin, or LPX1, then stimulated with different TLR ligands. ** *p* < 0.001 *** *p* < 0.0001 unpaired, non-parametric *t*-test (Mann–Whitney test).

**Figure 3 ijms-26-03830-f003:**
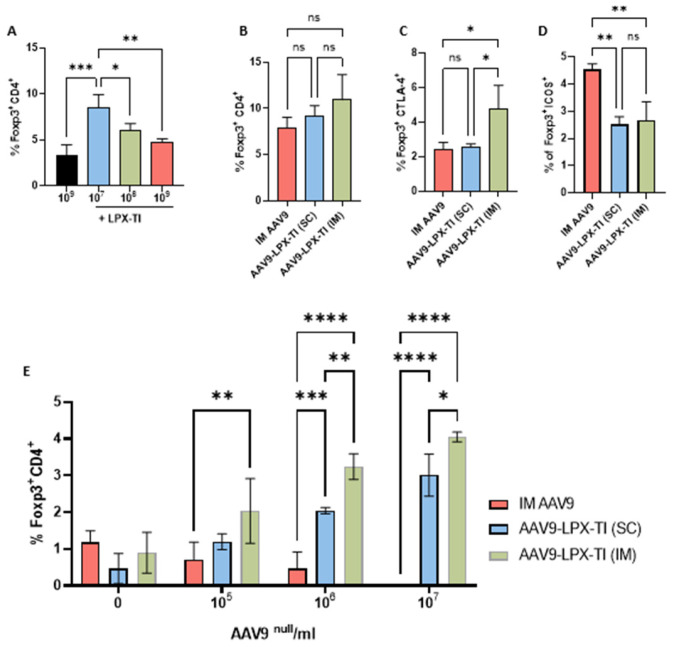
(**A**) Percentage of Treg expression in splenocytes treated with AAV9^null^-LPX-TI and different doses of AAV9^null^. (**B**) Treg expression in splenocytes from mice that received LPX-TI via subcutaneous (SC) or intramuscular (IM) injection. (**C**,**D**) Expression of CTLA-4 and ICOS in Tregs from splenocytes of mice treated with LPX-TI via SC or IM injection. (**E**) Foxp3 expression in splenocytes after recall response in mice treated with AAV9^null^-LPX-TI via SC or IM injection. * *p* < 0.05, ** *p* < 0.001, *** *p* < 0.0001, **** *p* < 0.00001, ns (not significant). All data presented as a means, error bars are SD.

**Figure 4 ijms-26-03830-f004:**
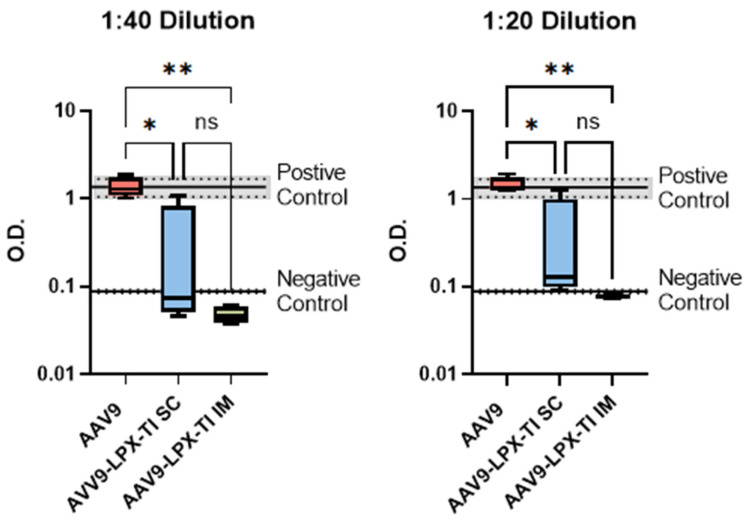
Anti-AAV9 antibody levels in the serum of mice treated with LPX-TI2/3 via s.c. or IM injection. * *p* < 0.05, ** *p* < 0.001, ns (not significant).

**Figure 5 ijms-26-03830-f005:**
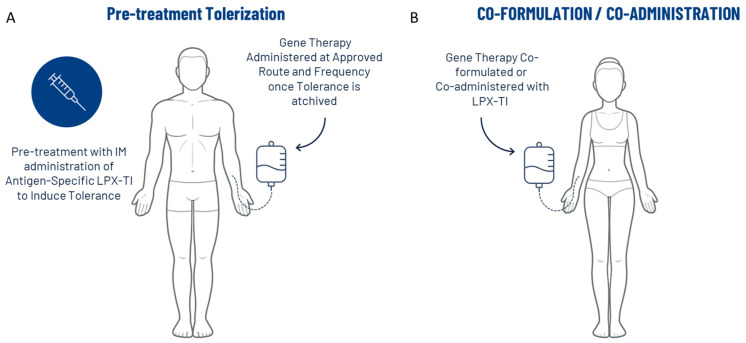
Schematic overview: proposed clinical application of immune tolerance induction using LPX-TI. (**A**) Suggested clinical application where immune tolerance to a therapeutic viral vector is induced by pre-treating the subject with null virus matching its serotype before administering the therapeutics dose at the intended route and dose. (**B**) Suggested clinical virus containing a genetic payload could be co-formulated with LPX-TI to induce tolerance upon administration.

**Table 1 ijms-26-03830-t001:** In silico binding affinity values of rigid docking of OPLS, LPX1, LPX2, and LPX3 to Tim.

Compound	OPLS	LPX1	LPX2	LPX3
Binding Affinity (kcal/mol)	−5.6 to −3.9	−6.0 to −4.9	−7.8 to −6.0	−5.2 to −4.5

## Data Availability

The data presented in this study are available on request from the corresponding author. (The data are not publicly available due to potential IP and confidentiality restrictions).

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
