# Peer review of "Tolerization with a Novel Dual-Acting Liposomal Tim Agonist Prepares the Immune System for the Success of Gene Therapy"

_ijms, 2025, doi:10.3390/ijms26083830_

Round 1

Reviewer 1 Report

Comments and Suggestions for Authors

A new strategy to reduce immunogenicity—a main obstacle in gene therapy, especially with adeno-associated viral vectors (AAV)—is investigated in the study, "Tolerization with a novel dual-acting liposomal Tim agonist prepares the immune system for the success of gene therapy." The authors want to create antigen-specific immunological tolerance by using first- and second-generation Tim receptor agonists (LPX1, LPX2, and LPX3) in a liposomal formulation that works in two ways. These formulations, particularly when encapsulating AAV9null, generate regulatory T cells (Tregs) and reduce immunological responses to the viral capsid proteins. The study shows that LPX1 is different from non-specific immunosuppressants like dexamethasone or rapamycin because it boosts Foxp3+CD4+ Tregs without really stopping innate immunity. LPX2 and LPX3 were created using computer modeling, and they bind more strongly to Tim receptors. They were then added to the liposomal membrane or hydration buffer to improve the tolerogenic response.

Most importantly, the in vivo studies indicate that intramuscular (IM) AAV9null/LPX-TI created a more robust population of tolerogenic Foxp3+/CTLA4+/ICOS^lo Tregs than subcutaneous (SC) delivery, therefore preventing the generation of anti-AAV9 antibodies. As shown by recall response studies, these Tregs were specific to antigens and functionally active, showing that they could tolerate the immune system for a long time. The technique could enable repeat dosing of gene therapy vectors, therefore addressing a major restriction in present treatments such as Zolgensma, in which patients develop antibodies preventing re-dosing. This method is seen as an improvement over earlier tolerance strategies, like antigen-fragment-loaded nanoparticles or immunosuppressant drugs, because it can be used with other viral vectors that were thrown out because they were too immunogenic.

Altogether, this work offers major progress in immune engineering for gene therapy. The authors present a possibly revolutionary solution to a long-standing clinical challenge by using dual-acting liposomal TIM agonists to establish persistent, antigen-specific immune tolerance, thereby overcoming vector immunogenicity to enable safe and repeatable gene therapy.

Author Response

We thank the reviewer for this thorough review. 

Reviewer 2 Report

Comments and Suggestions for Authors

The manuscript, "Tolerization with a Novel Dual-Acting Liposomal Tim Agonist Prepares the Immune System for the Success of Gene Therapy," is an original work focused on evaluating a novel dual-acting liposomal formulation that induces immune tolerance toward adeno-associated virus 9 (AAV9). The authors present in silico, in vitro, and in vivo (mouse model) experiments supporting the potential utility of this chemical formulation.

The issue of repeated dosing in viral-based gene therapy is critical, as immune responses often hinder the efficacy of long-term treatments. Therefore, improving viral-based gene therapy formulations is a highly relevant area in advanced medicine. This study is particularly appealing, as viral vectors remain the most effective platforms for delivering therapeutic nucleic acids into cells. The manuscript is well-written and well-organized, presenting a body of evidence that supports the authors' hypothesis.

General Comments

I would appreciate further discussion regarding the potential clinical applications of this novel formulation. For example, if available, data on the in vivo toxicity profile of the lipid molecules should be included. Additionally, how does this formulation impact the transduction efficiency of AAV9 in target tissues?

Another question arises regarding the choice of an empty AAV9 vector for the experiment rather than a reporter AAV9 expressing GFP or a similar marker protein. The balance between safety and efficacy is a key aspect of gene therapy. Therefore, the authors should reflect on these concerns and clarify how their experimental design addresses—or overlooks—certain critical questions that still need to be answered.

Specific Comments

  • Line 26: More gene therapies have been approved by the FDA. Please update this information by consulting the following website:
    https://www.fda.gov/vaccines-blood-biologics/cellular-gene-therapy-products/approved-cellular-and-gene-therapy-products
  • Line 28: Some viruses used in gene therapy, such as lentiviruses, are enveloped viruses. Given this, it would be more appropriate to refer to "protein or lipid-protein capsids"
  • Line 118: Numerical values should be correctly formatted using superscripts where necessary. This applies to other parts of the manuscript as well.
  • Line 225: More details regarding the bioinformatics tools and docking procedures used in the assays are required.
  • Line 250: Additional information on the quality control of AAV stocks is needed. This should include data on integrity (e.g., electron microscopy), titers expressed in genome copies (qPCR), and transduction potency (e.g., in vitro assays). If these analyses were conducted by the AAV provider, this should be explicitly stated to confirm that the stocks used were of sufficient quality for in vivo administration
  • Line 253: How many mice were used in the experiments? This is a crucial detail that must be included. Furthermore, a paragraph on the statistical analysis of the data should be added to the Materials and Methods

Author Response

General Comments

I would appreciate further discussion regarding the potential clinical applications of this novel formulation. For example, if available, data on the in vivo toxicity profile of the lipid molecules should be included. Additionally, how does this formulation impact the transduction efficiency of AAV9 in target tissues?

Response:

We have expanded the discussion section and added a figure to help the reader envision the clinical applications. We also acknowledged that we do not yet know the effect on transduction efficiency. The following language has been added:

"We propose two possible scenarios for the clinical application of this approach: First induce immune tolerance to therapeutic viral vector by pre-treating the subject with null virus matching its serotype before administering the therapeutics dose at the intended route and dose (Fig. 5A). This aligns with data presented here using AAV9null virus. Second, the virus containing a genetic payload could be co-formulated with LPX-TI to induce tolerance upon administration (Fig. 5B) with or without pre-tolerization. We did not evaluate this option in the current report; however, we plan to address this in future studies designed to evaluate both tolerization towards the envelope/capsid as well as the transduction efficacy of gene therapy coformulation with LPX-TI. "

Another question arises regarding the choice of an empty AAV9 vector for the experiment rather than a reporter AAV9 expressing GFP or a similar marker protein. The balance between safety and efficacy is a key aspect of gene therapy. Therefore, the authors should reflect on these concerns and clarify how their experimental design addresses—or overlooks—certain critical questions that still need to be answered.

Response:

As shown above, the use of null virus is valid to the first envisioned clinical application. The second option will require the use of viral vector with genetic payload and will be explored as the program advances.  

Specific Comments

  • Line 26: More gene therapies have been approved by the FDA. Please update this information by consulting the following website:
    https://www.fda.gov/vaccines-blood-biologics/cellular-gene-therapy-products/approved-cellular-and-gene-therapy-products

Response:

Thank you for the reference. we have updated the number of approval and added the link as a reference.

  • Line 28: Some viruses used in gene therapy, such as lentiviruses, are enveloped viruses. Given this, it would be more appropriate to refer to "protein or lipid-protein capsids"

Response

The now reads: "Viral vectors comprising of protein or lipid-protein"

  • Line 118: Numerical values should be correctly formatted using superscripts where necessary. This applies to other parts of the manuscript as well.

Response

Thank you for pointing this out. This has been corrected throughout the manuscript 

  • Line 225: More details regarding the bioinformatics tools and docking procedures used in the assays are required.

Response:

This section has been expanded substantially. The following has been added:

"Docking Simulations: AutoDock Vina (ADVina) was used to perform docking simulations in order to assess the binding affinities of several proposed molecular structures to the Tim receptor.

Preparation for Docking: To prepare for docking, the co-crystal structure of the Tim- protein bound to its endogenous ligand, dicaproyl-phosphatidylserine (PS), was downloaded from the Protein Data Bank (PDB ID: 3BIB). The ligand (PS) was removed from the binding site to allow docking of the proposed molecules. All water molecules were removed, except for one, which was essential for forming a hydrogen bond bridge between PS and the protein. Charges for all atoms in the protein were calculated at a physiological pH of 7.4, matching the conditions used in the in vitro experiments. Similarly, charges for all docked molecules were also computed at the same physiological pH. Docking was performed with both rigid and flexible protocols, and the top 20 binding poses for each molecule were identified for further analysis.

Validation of the Docking Method: To validate the docking approach, the endogenous ligand (PS) was re-docked into the protein’s binding site after its removal. The root mean square deviation (RMSD) of the pose the matched the native configuration was calculated and ranked to demonstrate the reliability of the docking procedure.

Docking of Experimentally Tested and Proposed Structures: After validating the docking method, several compounds—OPLS, LPX1, LPX2, and LPX3—were docked into the Tim receptor’s binding site using both rigid and flexible docking methods. These results were compared with experimental data"

  • Line 250: Additional information on the quality control of AAV stocks is needed. This should include data on integrity (e.g., electron microscopy), titers expressed in genome copies (qPCR), and transduction potency (e.g., in vitro assays). If these analyses were conducted by the AAV provider, this should be explicitly stated to confirm that the stocks used were of sufficient quality for in vivo administration

Response: 

The following had been added to the material and method section:

"AAV9null were sourced from Vigene Biosciences catalog number RS-AAV9-ET. The physical particle concentration as determined by ELSA was 1.79X1012 VP/ml. Purity, size and stoichiometry of the viral protein was determined by reduced SDS-PAGE. Sourced AAV9null was negative for mycoplasma, endotoxin level was <10 EU/ml as determined by LAL test. All tests were performed by Vigene Biosciences per their SOPs."  

  • Line 253: How many mice were used in the experiments? This is a crucial detail that must be included. Furthermore, a paragraph on the statistical analysis of the data should be added to the Materials and Methods

Response: Number of animals has been added. Statistics is now covered by the following paragraph:

"Statistics: For titer analysis in Fig 2A, unpaired, non-parametric t-test (mann-whitney test) was used. 2-way ANOVA with multiple comparison to the control was used for Fig 2C and 3E. Everywhere else, one-way ANOVA was used. All statistical analysis was done using GraphPad Prism 10

Round 2

Reviewer 2 Report

Comments and Suggestions for Authors

The authors have adequately addressed all my concerns. I have no further concerns about the study.